

# Touch-screen-guided task reveals a prosocial choice tendency by chimpanzees (*Pan troglodytes*)

Renata S. Mendonça[1,2,*], Christoph D. Dahl[3,*], Susana Carvalho[2,4,5,6], Tetsuro Matsuzawa[7] and Ikuma Adachi[8]

[1] Primate Research Institute, Section of Language and Intelligence, Kyoto University, Inuyama, Japan
[2] Centre for Functional Ecology—Science for People & the Planet, Department of Life Sciences, University of Coimbra, Coimbra, Portugal
[3] Institute of Biology, Department of Comparative Cognition, University of Neuchatel, Neuchâtel, NE, Switzerland
[4] Primate Models for Behavioural Evolution Lab, Institute of Cognitive and Evolutionary Anthropology, University of Oxford, Oxford, United Kingdom
[5] Interdisciplinary Centre for Archaeology and Evolution of Human Behaviour (ICArEHB), University of Algarve, Faro, Portugal
[6] Gorongosa Restoration Project, Gorongosa National Park, Sofala, Mozambique
[7] Institute for Advanced Studies, University of Kyoto, Kyoto, Japan
[8] Primate Research Institute, Center for International Collaboration and Advanced Studies in Primatology, Kyoto University, Inuyama, Japan

[*] These authors contributed equally to this work.

Corresponding author
Renata S. Mendonça,
mendonca.renata.8m@kyoto-u.ac.jp,
renatadasilva.mendonca@gmail.com

## ABSTRACT

Humans help others even without direct benefit for themselves. However, the nature of altruistic (i.e., only the other benefits) and prosocial (i.e., self and other both benefit) behaviors in our closest living relative, the chimpanzee, remains controversial. To address this further, we developed a touch-screen-guided task that allowed us to increase the number of trials for a thorough test of chimpanzees' prosocial and altruistic tendencies. Mother-offspring dyads were tested in the same compartment; one was the actor while the other was the recipient. In Experiment 1, the actor chose among three options: prosocial, selfish (only the actor benefited) and altruistic. To better understand the nature of the chimpanzees' choices and to improve experimental control, we conducted two additional experiments. Experiment 2 consisted of two-option choices interspersed with three-option choices, and in Experiment 3 the two-option choice were blocked across all trials. The results of Experiment 1 clearly showed that chimpanzees acted prosocially in the touch-screen-guided task, choosing the prosocial option on an average of 79% of choices. Five out of the six chimpanzees showed the preference to act prosocially against chance level. The preference for the prosocial option persisted when conditions were changed in Experiments 2 and 3. When only selfish and altruistic options were available in Experiments 2 and 3, chimpanzees preferred the selfish option. These results suggest that (1) most individuals understood the nature of the task and modified their behavior according to the available options, (2) five out of the six chimpanzees chose to act prosocially when they had the option to, and (3) offspring counterbalanced between altruistic and selfish, when given those two options perhaps to avoid suffering repercussions from the mother.

# INTRODUCTION

Humans are clearly a case in which social exchange increases the relative fitness of individuals who engage in altruistic behaviors, enabling altruism to diffuse through subsequent generations (*Fehr & Fischbacher, 2003*; *Warneken & Tomasello, 2009*). Humans frequently help others without directly benefiting themselves (*Fehr & Gächter, 2002*; *Fehr & Fischbacher, 2004*). Prosocial behavior is described as any behavior that includes actions intended to benefit another, such as helping, comforting, sharing resources and cooperating (*Batson & Powell, 2003*). Altruism is a motivational concept in which the actor does not consciously regard his self-interests (*Hoffman, 1978*; *Batson & Powell, 2003*). Therefore, this behavior can benefit the recipient while entailing costs to the actor, or in the absence of any obvious proximate reward (*Batson & Powell, 2003*; *De Waal, 2008*). This concept contrasts with egoism (here referred to as selfish behavior), which has the ultimate goal of increasing one's own welfare (*Mueller, 1986*). How did prosocial behaviors evolve in humans? Comparative studies can provide important perspectives for addressing this question. In recent decades multiple studies have explored prosocial and altruistic behaviors in nonhuman primate species (*De Waal, 2008*; *Lakshminarayanan & Santos, 2008*; *Cronin, Schroeder & Snowdon, 2010*; *Skerry, Sheskin & Santos, 2011*; *Horner et al., 2011a*; *Takimoto & Fujita, 2011*; *Suchak & De Waal, 2012*; *Kim et al., 2015*). To understand the mechanisms that underlie prosocial and altruistic behavior, the chimpanzee is a good model for the following three major reasons: (1) they share a recent common ancestry with humans, which makes them a good comparative model for studying the evolution of human behavior (*McGrew, 2010*); (2) some observational studies have reported prosocial behavior in chimpanzees (*Nishida & Hosaka, 1996*; *Watts, 1998*; *Langergraber, Mitani & Vigilant, 2007*; *Crockford et al., 2012*); and (3) empirical evidence shows that chimpanzees understand other individuals' intentions (*Hare, Call & Tomasello, 2001*; *Yamamoto, Humle & Tanaka, 2012*).

Chimpanzees' cooperative and prosocial tendencies have been studied in a range of settings (*Hirata, 2009*). However, the issue of prosociality remains controversial, as some studies have failed to show such tendencies (*Silk et al., 2005*; *Jensen et al., 2006*; *Vonk et al., 2008*) and other reported prosocial tendencies only slightly above chance level (*Warneken et al., 2007*; *Horner et al., 2011a*; *Melis, Schneider & Tomasello, 2011*; *Melis et al., 2011*; *Claidière et al., 2015*). Two main experimental paradigms have been used to test prosociality in non-human primates (*Horner et al., 2011a*), namely using: (1) giving assistance tests, in which the subject has to choose between helping, by providing instrumental help, or not helping the recipient and (2) prosocial choices tests (PCT), in which the subject has to choose between a prosocial (allowing subject and recipient to be rewarded) or selfish option (only the subject is rewarded). Some PCT studies have failed to show a clear prosocial preferences in chimpanzees (*Silk et al., 2005*; *Jensen et al., 2006*), arguably due to methodological constraints. Even with improved paradigms, results are unclear (60% prosocial) (*Horner et al., 2011a*) and open to challenge (*Skoyles, 2011*), given the frequent selection (40%) of selfish tokens, when a choice between selfish and prosocial tokens was presented by the experimenter. However, authors have argued that organisms do not

choose categorically but rather sample the choices from time to time, which may result in a high proportion of selfish choices (*Horner et al., 2011b*).

We developed a touch-screen-guided task to re-examine the existence of prosocial and altruistic behaviors, as well as the factors modulating their choices, using a new paradigm. We tested three mother-offspring pairs of chimpanzees who had experience with various computer-controlled experiments (*Matsuzawa, 2003*; *Matsuzawa, 2006*; *Martin et al., 2014*). Unlike most of the prosociality studies (but see: *House et al., 2014*; *Suchak et al., 2014*; *Claidière et al., 2015*), we tested the actor and recipient individuals in the same compartment and we used a touch-screen-guided procedure that allowed us to increase the number of trials per individual. A prosocial option was defined as the chimpanzee playing the role of actor choosing to reward both actor and recipient. An altruistic option was defined as the act of providing reward only to the recipient. A selfish option was defined as the actor choosing to reward only himself. We ran three experiments to examine how prosocial, selfish and altruistic tendencies were modulated across different conditions. In Experiment 1, chimpanzees were requested to choose among prosocial (P), selfish (S) and altruistic (A) options. In Experiments 2 and 3 they were given two of the three options. Experiment 2 consisted of choosing between two out of three choices that were presented randomly across the trials. Experiment 3 consisted of three sessions, each one with two out three choices (for example, one session only with prosocial and altruistic options, another with altruistic and selfish, and another with prosocial and selfish) blocked across the trials.

We hypothesized that chimpanzees show a tendency to behave prosocially (above selfishly and altruistically), and this tendency varies according different conditions. The following predictions were formulated for each condition/experiment: experiment (1) if chimpanzees have a tendency to behave prosocial, they should choose the prosocial option more often than the selfish and altruistic options; experiment (2) if chimpanzees understand the meaning of the outcomes (a) they should keep their prosocial preference and (b) when given a choice between two out of the three options, they should show a preference for one of the options; experiment (3) if chimpanzees are presented with two out of the three options constantly across the trials, they may counterbalance their choices to avoid repercussions from other individuals.

## GENERAL METHODS

### Participants

Six chimpanzees (*Pan troglodytes*): one juvenile male (Ayumu, 12 years old), two juvenile females (Cleo and Pal, around 12 years of age) and three adult females (Ai, Chloe, and Pan, all around 30 years of age) participated as mother-offspring pairs. Because of their mother-offspring relationship individuals had to be tested in the same compartment: Ai with Ayumu (Am), Chloe (Ch) with Cleo (Cl), Pan (Pn) with Pal (Pl) (Fig. 1). The chimpanzees live in groups of six and seven individuals in indoor-outdoor enclosures at the Primate Research Institute, Kyoto University. The outdoor enclosure (770 m$^2$) is environmentally enriched with artificial streams containing fish and more than 400 species of plants, in addition to ropes and climbing structures up to 15 m high, and has direct access

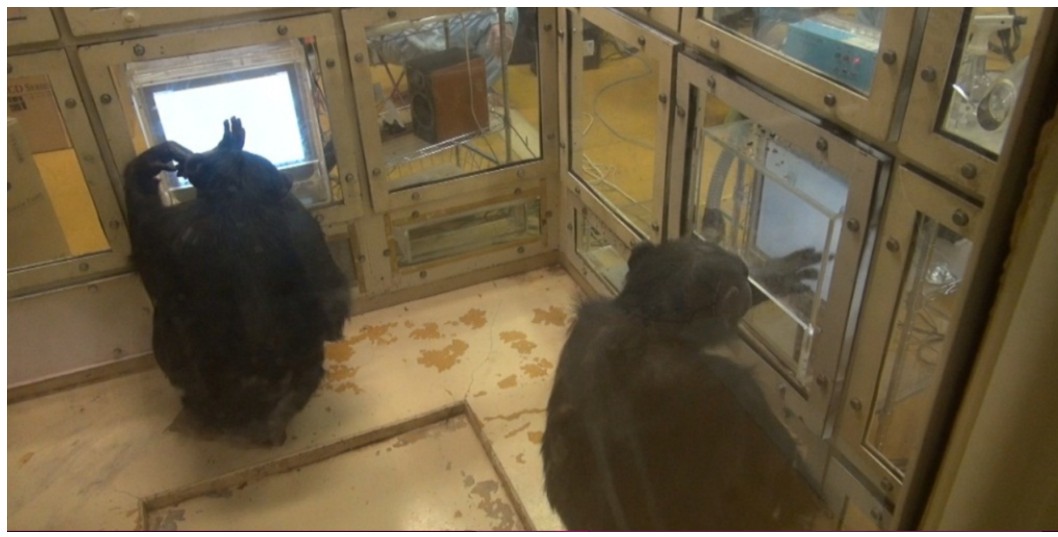

**Figure 1** **Pan (Pn) and Pal (Pl) performing the task and sharing the same compartment.** Pan as the actor on the right and Pal as the recipient on the left. Photo credit: Renata Mendonça.

to indoor quarters. All subjects had previously participated in various computer-controlled perceptual–cognitive experiments (*Matsuzawa, 2003*; *Matsuzawa, 2006*; *Adachi, 2014*) including some in similar social settings (*Martin, Biro & Matsuzawa, 2011*; *Martin et al., 2014*).

## Apparatus

We used two 17-inch LCD touch panel displays (1,280 × 1,024 pixels) controlled by custom-written software under Visual Basic 2010 (Microsoft Corporation, Redmond, Washington, USA). Chimpanzees sat in one experimental chamber (approximately 2.5 m wide, 2.5 m deep, 2.1 m high), while the experimenters sat outside the booth, separated from the chimpanzees by transparent acrylic panels (Fig. 1). The displays were placed into the acrylic panels. The appropriate distance between the active subject and its display was 40 to 50 cm. Options appeared on the screen in sizes of about three to four degrees of visual angle. The subjects responded by touching the options on the display surface with a finger. A transparent acrylic panel fitted with an opening allowed manual contact with the display while protecting it from damage. A food tray was installed below each display, for delivering food rewards via a universal feeder (Bio Medica BUF-310P50). Displays and feeders were automatically controlled by the same program that controlled the display of the stimuli.

## Stimuli

To initiate the task, a circular button was presented as stimuli in the bottom of the actor's screen. After pressing the start key, three grayscale 3-D shape options (cube, cylinder and sphere) horizontally aligned with equal spacing on the computer monitor of one of the two chimpanzees (Fig. 2). Each symbol represented each given option: altruistic, prosocial and selfish. To facilitate the association of the options with their corresponding

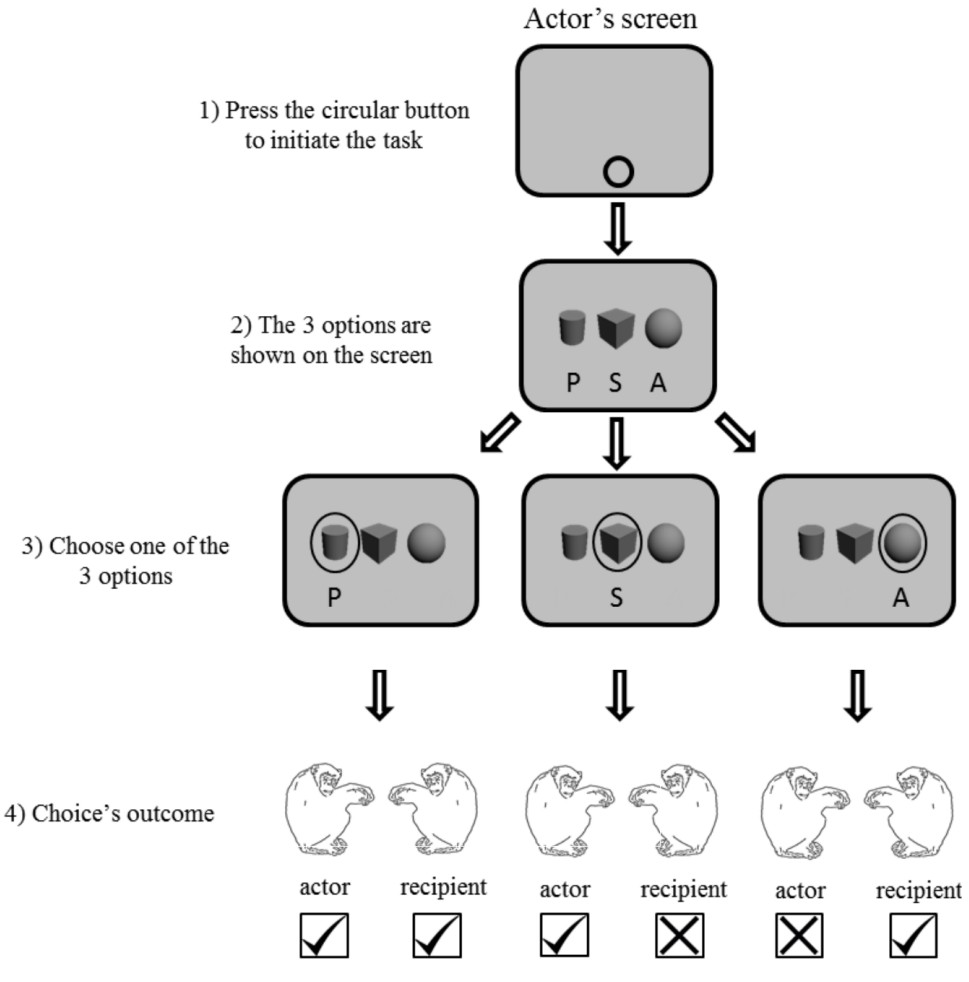

Figure 2 **Procedure of Experiment 1 "three choices condition".** Grey screen represents the screen of the actor.

function, the position of the stimuli was fixed for each participant but counterbalanced across participants. In Experiment 1, the three options were presented simultaneously on the actor's screen. In Experiments 2 and 3, two out of the three options were presented. In Experiment 2, two-option choice trials (two out of three options) were interspersed with three-option choice trials (as in Experiment 1). In the two-option choice trials, the combination of options was randomly assigned across the individuals. In Experiment 3, one of the three possible combinations of two options was constant across the block of trials. Therefore, we ran three different sessions, each one with two options (out of the three) available across trials (Table 1). The monitor of the second chimpanzee showed a mid-grey blank screen throughout the sessions.

**Table 1** Summary of the description of each experiment.

| Experimental condition | Description of the task |
| --- | --- |
| Experiment 1 | Three choices were presented - prosocial (P), selfish (S), altruistic (A) for three sessions of 48 trials, totalling 144 trials (Fig. 2). |
| Experiment 2 | Two out of three choices (P-S, S-A, P-A) were presented randomly, interspersed with three choice trials (P, S, A) at a rate of 1:5. We ran three sessions of 32 trials. |
| Experiment 3 | Two out of three choices were presented. We ran 96 trials for each pair of choices (P-S, S-A, P-A). Each pair was presented in a block of 48 trials and the order of choices counterbalanced between individuals. |

## Procedure

### Training and learning phase

In the training phase, the individuals were trained to discriminate three different sounds corresponding to the three different outcomes (prosocial, selfish and altruistic). The sounds used in this experiment were unfamiliar to chimpanzees, so they would not associate with the regular rewarding sound used with other touchscreen tasks. In the training phase, the individuals were tested alone. The actor was placed in in front of his/her monitor, and the recipient was separated in the other compartment of the booth. We did not request the actor to touch the screen in this phase. The actor had access to both feeders, including the recipient feeder. The actor could easily hear the sound and pick up the reward on the recipient feeder. This procedure should ensure that the individuals understand that both feeders provided food. We randomly chose which sounds to play paired with the location of the outcome. We ran 200 trials for each individual.

In the learning phases, we trained the chimpanzees to associate the assigned shape with their respective function. Both chimpanzees were now placed in the same compartment, in front of their respective monitors (Fig. 1). We ran three sessions, in which only one of the three stimuli was presented for 24 trials: 24 trials with the presentation of the cube, 24 trials with the sphere and 24 trials with the cylinder. Shapes' functions were randomly assigned across the subjects.

### Experimental phase

In the experimental phase, chimpanzees were tested in actor-recipient pairs, in the same compartment of the experimental booth (Fig. 1), approximately 0.40 m apart. One degree of gaze angle corresponded to approximately 0.7 cm on the screen at a viewing distance of 40 cm. One chimpanzee was the actor while the other was the recipient; role was randomly assigned across sessions. Each trial was initiated by the actor pressing a green button on the middle bottom of screen. The actor made a choice by touching one of the three options presented on the screen. A food reward (an apple piece, approximately 1 cm$^3$) was given according to the assignments of the options and their functions. The three options consisted in rewarding only the actor (selfish (S)), both participants (prosocial (P)) or only the recipient (altruistic (A)). Feeder activation was accompanied by two distinctive buzzer sounds with slight temporal delay to indicate clearly which feeder was giving the reward. Throughout the procedure the recipient sat in front of a grey screen. After the completion

of the experiment (by the end of the third session) the chimpanzees changed positions: the actor moved to the recipient's place and vice-versa. Each pair received three sessions for each role, totaling six sessions per day.

In Experiment 1, we ran three sessions of 48 trials (144 trials in total) for each actor. The actor could choose among three options on the screen: P, S and A (Fig. 2). In Experiment 2, option assignments and locations on the screen were as in Experiment 1, except that we also reduced the number of options from three to two: prosocial and selfish (P-S), selfish and altruistic (A-S), or prosocial and altruistic (P-A) (Table 1). We ran 32 trials of each combination, giving a total of 96 trials for each subject. These two-option trials were randomly interspersed with three-option trials at a ratio of 1:5, to ensure that chimpanzees could associate this new condition with the previous one, as the conditions have been conducted in different days. However, because our focus was on the two-option trials, we only analysed those trials in this study. By reducing the options in some of the trials, we turned the social event into a more critical decision-making situation than in Experiment 1 (three-option-choices) and, hence, increased the social pressure between partners and possible repercussions toward the active partner.

In Experiment 3, to further explore the dynamics of the two-option task and increase the social pressure between the partners, we provided each of the two option choices in blocks of 48 trials in a counterbalanced order across participants (Table 1). Experiment 3 involved presentations of two options at the same time and consisted of 96 trials presented in two sessions for each combination of two trial types: P-S, A-S, P-A.

All experiments were carried out in accordance with the 2002 version of the Guidelines for the Care and Use of Laboratory Primates by the Primate Research Institute, Kyoto University. The experimental protocol was approved by the Animal Welfare and Care Committee of the same institute (protocol# 2012-090).

### Data analysis

Data analysis was performed using R 3.3.1 (*R Core Team, 2015*) in R-studio 0.99.463 (*RStudio Team, 2015*). For individual testing, we used Chi-square tests for the three-choice experiment (Experiment 1) and binomial tests for the two-choice experiment, (Experiment 2 and 3), to examine subjects' performance against chance level. We rejected the null hypothesis if $P < 0.05$. Additionally, we use the function geom_smooth, method = "loess" from the package "ggplot2" to fit a line using linear smoothing for the figures corresponding to each experiment. The curve given by geom_smooth function produces an estimate of the conditional mean function. The shaded band represents a pointwise 95% confidence interval on the fitted values (given by the line).

## RESULTS

### Experiment 1

Five out of the six individuals chose the prosocial option above chance level (Chi-squared, Table 2). One of the six individuals (Pn) showed the opposite trend and preferentially choose the selfish option more often than prosocial, and this tendency increased across the trials.
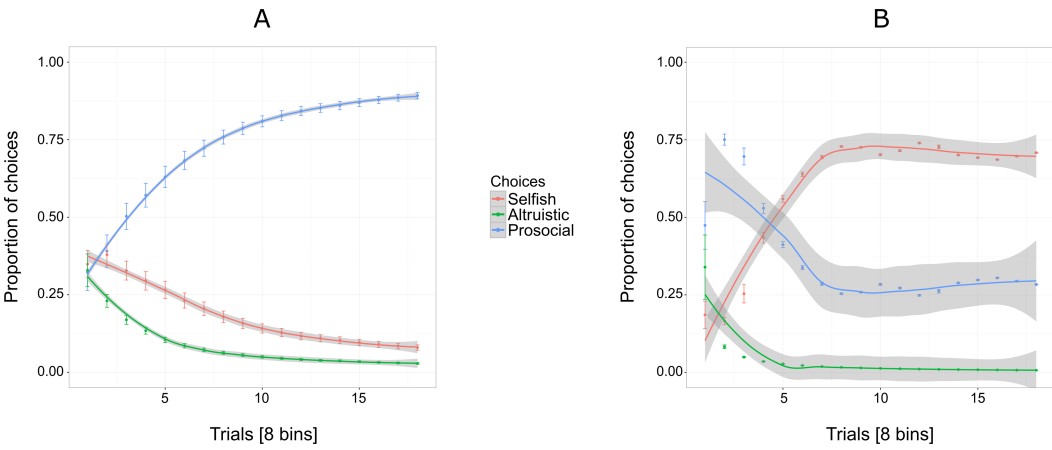

**Figure 3** **Proportion of the cumulative mean in Experiment 1 for selfish, altruistic and prosocial options as a function of trials ($x$-axis) for five individuals (A) and Pan (B).** Trials are grouped in bins (each comprising eight trials) for a total of 144 trials, i.e., 18 bins. Error bars represent standard errors of the means. The shaded band represents the pointwise 95% confidence interval on the fitted value.

**Table 2** **Results of the chi-square test for all the individuals in Experiment 1.**

| Individuals | Age class | Prosocial choice | Selfish choice | Altruistic choice | Chi-square | *P*-value |
|---|---|---|---|---|---|---|
| Ai | Mother | 0.82 | 0.15 | 0.03 | 155.79 | <0.001 |
| Am | Offspring | 0.84 | 0.13 | 0.03 | 168.29 | <0.001 |
| Ch | Mother | 0.91 | 0.05 | 0.03 | 215.38 | <0.001 |
| Cl | Offspring | 1 | 0 | 0 | 288 | <0.001 |
| Pn | Mother | 0.29 | 0.70 | 0.01 | 105.29 | <0.001 |
| Pl | Offspring | 0.9 | 0.06 | 0.03 | 210.29 | <0.001 |

Figure 3A shows an increasing overall preference for the prosocial option over all 144 trials for five out of six individuals, while the preference for the selfish and altruistic options decreased over the trials. The selfish individual, Pn was plotted separately (Fig. 3B) to show her preference for the selfish option over the prosocial and altruistic across trials.

Experiment 1 reveals an exploratory phase, in which, in the beginning individuals (except for Cl) were choosing the three options at similar proportions (first bin of eight trials, Fig. 3A) until they started showing a preference for the prosocial option with the increase of trials.

## Experiment 2

Four of the six individuals chose the prosocial option above chance level (Fig. 4A). Am did not choose the prosocial option above chance level in this experiment (Binomial test, Table 3). The selfish subject, Pn, kept choosing the selfish option more than the prosocial option, as she did in Experiment 1, thereby deviating from the pattern shown by the other participants (Fig. 4B). In this experiment, the proportion of prosocial choices, for three out of the six individuals (Am, Ch and Pn), decreased in this experiment, compared to

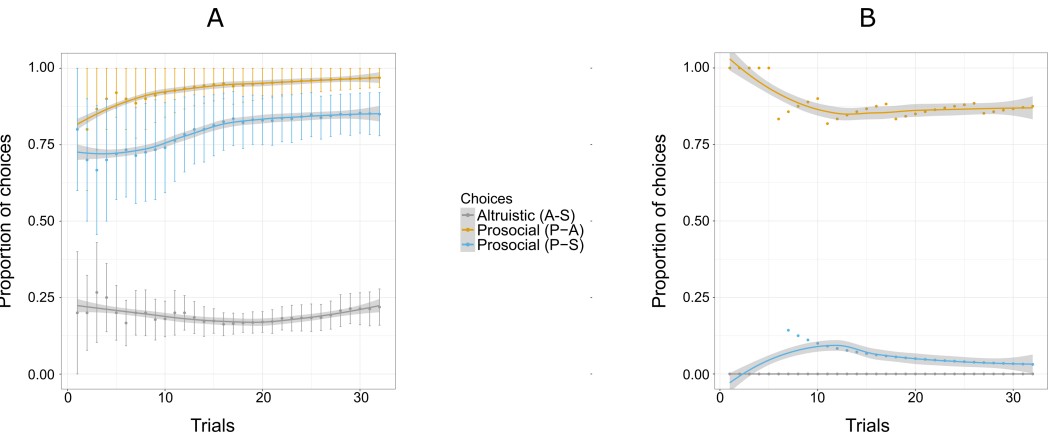

**Figure 4** **Proportion of the cumulative mean for combinations of two options: S-A, (selfish and altruistic), P-A (prosocial and altruistic) and P-S (prosocial and selfish) in Experiment 2 for five individuals (A) and Pan (B).** "Altruistic (A-S)" means choosing the altruistic option when the altruistic and the selfish options are given; "Prosocial (P-A)" means choosing the prosocial option when the prosocial and the altruistic choices are given; "Prosocial (P-S)" means choosing the prosocial option when the prosocial and the selfish options are given. Error bars represent standard errors of the means (A). The shaded band represents the pointwise 95% confidence interval on the fitted value.

**Table 3** **Results of the binomial test for all the individuals in Experiment 2.**

| Individuals | Age class | Prosocial choice over selfish | *P*-value | Prosocial choice over altruistic | *P*-value | Selfish choice over altruistic | *P*-value |
|---|---|---|---|---|---|---|---|
| Ai | Mother | 0.90 | <0.001 | 1 | <0.001 | 0.94 | <0.001 |
| Am | Offspring | 0.62 | 0.21 | 1 | <0.001 | 0.75 | 0.007 |
| Ch | Mother | 0.75 | 0.007 | 0.84 | <0.001 | 0.88 | <0.001 |
| Cl | Offspring | 1 | <0.001 | 1 | <0.001 | 0.50 | 0.378 |
| Pn | Mother | 0.03 | <0.001 | 0.88 | <0.001 | 1 | <0.001 |
| Pl | Offspring | 0.97 | <0.001 | 1 | <0.001 | 0.75 | 0.007 |

Experiment 1. The probability of mothers choosing the selfish option over the altruistic and prosocial options increased in Experiment 2 compared to Experiment 1 (Fig. 5). Unlike in Experiment 1, the chimpanzees' choices did not vary over trials suggesting that individuals may have remembered the symbol assignments from the previous experiment.

## Experiment 3
Similar to Experiment 1, the individual's responses showed that five of the six individuals chose the prosocial option above chance level (Binomial test, Table 4). Similar to Experiment 2, the proportion of choosing the prosocial key was kept constant across the trials, for five out of six individuals (Fig. 6A). Like in Experiments 1 and 2, Pn stood out from other participants by choosing the selfish option over the prosocial option (Fig. 6B). Overall, the proportion of prosocial choices over selfish increased from the Experiment 2 for three individuals (Ai, Am and Ch).

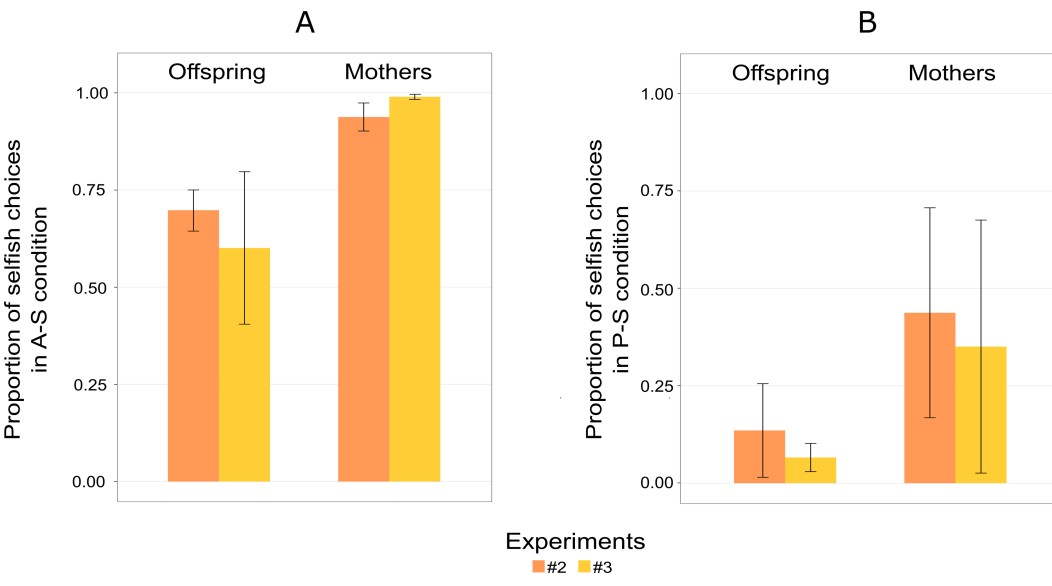

**Figure 5 Mean proportion of choices for S-A (selfish and altruistic) trials (A), and mean proportion of choices for S-P (selfish and prosocial) trials (B) split into offspring and mothers for Experiment 2 and 3.** "Selsfish (A-S)" means choosing the selfish option when the altruistic and the selfish options are given (A); "Selfish (P-S)" means choosing the selfish option when the prosocial and the selfish choices are given (B). Error bars are standard errors of the means.

**Table 4 Results of the binomial test for all the individuals in Experiment 3.**

| Individuals | Age class | Prosocial choice over selfish | *P*-value | Prosocial choice over altruistic | *P*-value | Selfish choice over altruistic | *P*-value |
|---|---|---|---|---|---|---|---|
| Ai | Mother | 0.99 | <0.001 | 1 | <0.001 | 0.99 | <0.001 |
| Am | Offspring | 0.93 | <0.001 | 1 | <0.001 | 0.98 | <0.001 |
| Ch | Mother | 0.96 | <0.001 | 1 | <0.001 | 0.98 | <0.001 |
| Cl | Offspring | 1 | <0.001 | 1 | <0.001 | 0.32 | <0.001 |
| Pn | Mother | 0.02 | <0.001 | 0.98 | <0.001 | 1 | 0.5 |
| Pl | Offspring | 0.88 | <0.001 | 0.96 | <0.001 | 0.5 | 1 |

Mothers showed a greater tendency than their offspring to choose the selfish option over the altruistic option (Fig. 5A). The probability of choosing the altruistic option also increased in Experiment 3, with the exception of the male offspring (Am), who showed a similar pattern to mothers (Table 4). Similar to Experiment 2, an exploratory phase was not observed.

## DISCUSSION

Overall, we found that prosocial behavior predominated over selfish and altruistic behaviors (Experiment 1). Prosocial responding was slightly more frequent when the alternative was altruistic responding (Experiments 2 and 3) compared to selfish, and all individuals show a clear preference for behaving prosocially over the altruistic option. In the early trials of

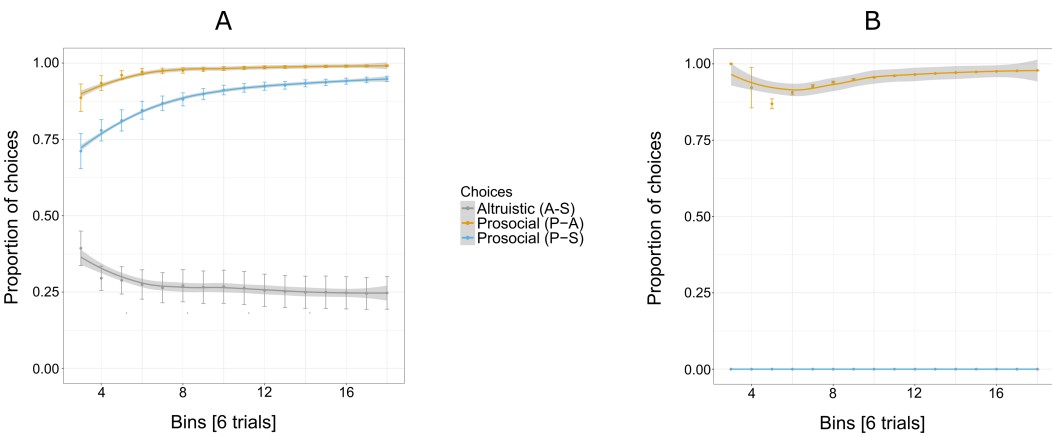

**Figure 6** Proportion of the cumulative mean for combinations of two options: S-A, (selfish and altruistic), P-A (prosocial and altruistic) and P-S (prosocial and selfish) in Experiment 3 for five individuals **(A)** and Pan **(B)**. "Altruistic (A-S)" means choosing the altruistic option when the altruistic and the selfish options are given; "Prosocial (P-A)" means choosing the prosocial option when the prosocial and the altruistic choices are given; "Prosocial (P-S)" means choosing the prosocial option when the prosocial and the selfish options are given. Trials are grouped in bins (each comprising six trials) for a total of 96 trials, i.e., 16 bins. Error bars represent standard errors of the means. The shaded band represents the pointwise 95% confidence interval on the fitted value.

Experiment 1, chimpanzees chose among the three options at close to the chance level; however, their options stabilized with increasing experience of the outcome of each choice. In Experiments 2 and 3 no such exploratory behavior was observed, suggesting that most of the chimpanzees (with exception of Cl) understood and remembered the outcome of their choices from Experiment 1. Cl always chose the prosocial option from the beginning of Experiment 1 and did not explore other outcomes. Therefore, there is the possibility that she may have just learned that the prosocial symbol was rewarding to her via simple associate learning or she may have learned to avoid the other choices.

Four out of the six chimpanzees showed a tendency towards prosocial behaviour, supporting findings of previous experimental studies (*Warneken et al., 2007*; *Horner et al., 2011a*; *Melis et al., 2011*) and evidence from observations in the wild (*Nishida & Hosaka, 1996*; *Watts, 1998*; *Duffy, Wrangham & Silk, 2007*). A potential limitation of the study by *Horner et al. (2011a)* concerns the low number of repetitions (30 trials). In the present study the proportion of prosocial choices made by the chimpanzees at around 30 trials (i.e., four bins) was similar to that in *Horner et al. (2011a)*. However, by increasing the number of trials (by a factor of 4.8) we increased the overall prosocial bias from an average of 60% in *Horner et al. (2011a)* to an average of 88%, and to 100% for five out of six chimpanzees. During the first phase of trials in Experiment 1, chimpanzees chose more equitably among the three options (exploratory phase), before eventually switching their preference for the prosocial option, a preference that persisted until the end of testing. The prosocial-selfish rate found in previous studies (e.g., *Silk et al., 2005*; *Jensen et al., 2006*; *Horner et al., 2011a*) may be, therefore, a consequence of subjects receiving fewer trials. In addition to experiencing more trials in the current experiment, it is also possible that having

the two individuals sharing the same compartment during the experiment motivated the chimpanzees to act more prosocially because of fear of repercussions from the partner. However, in the study of *Claidière et al. (2015)* which tested chimpanzee pairs in the same and in separate adjacent enclosures found that chimpanzees behave more prosocially when they were separated.

*Tennie, Jensen & Call (2016)* have shown that chimpanzees' willingness to help others may depend on the experimental settings, therefore prosociality could arise as a by-product of the experimental design. Further studies are required to better address this question, as we could not control for the effect of sharing the same chamber in these experiments.

In contrast to our results, no modulation of prosocial behavior by relative social rank was observed in *Horner et al. (2011a)*. It can be argued that the lack of any rank-related influence on prosocial behavior might be due to the physical separation of the two actors in that study. The fear of potential repercussions from the mothers could explain why the female offspring acted more altruistically (given the selfish option) compared to their mothers in Experiment 3.

One may argue that chimpanzees were choosing the prosocial option with the intention of scrounging the reward from the partner; however, we did not observe any scrounging behavior or attempt to steal the reward during the experiment. Moreover, we also did not observe any signs of frustration by the partner, when they were most likely to occur, in Experiment 3, when given the choice between acting selfishly or altruistically.

Some previous studies that failed to show, or showed little evidence of prosociality appear more complex methodologically and may have require extra cognitive effort compared to the task used in our study. Examples include using tokens to exchange for food rewards with a human experimenter (e.g., *Horner et al., 2011a*), or using a stick as a tool to dislodge food rewards (e.g., *Vonk et al., 2008*). In those cases, actors behaved ''prosocially'' even in a ghost condition in which no conspecific was present. Given the settings of our experiment, we could not run a condition with the partner being absent. If we had run the ghost condition, the actor could try to maximize the reward by choosing the prosocial option in the absence of a partner, thus spoiling the association between the key and the reward outcome. If we had blocked the passage of the recipient, we would have to run the experiment with both subjects separated from the beginning which was not our goal, as we wanted to test individuals in the same compartment to increase social pressure. Therefore, to be able to run a ghost condition we would have to change our settings from the start. Further experiments should take these matters into account.

Although rank turned out to be an important factor in our study (with mothers being the more dominant individuals), because we tested only mother-offspring pairs, we could not examine the influence of kinship separate from rank. Considering the various differences we found in the response patterns between the mothers and their respective offspring, we cannot support the suggestion that chimpanzees return past favors (*Gomes, Mundry & Boesch, 2009*; *Gomes & Boesch, 2011*). As stated in *Horner et al. (2011a)*, this lack of evidence might be related to the fact that cooperative behaviors such as hunting (*Boesch & Boesch, 1989*; *Boesch, 1994*) patrolling and coalitions (*Mitani, Merriwether & Zhang, 2000*)

are more typical of male than female chimpanzees. We tested five females and only one male; clearly, further studies are needed to address the question of sex differences regarding prosocial tendencies.

One chimpanzee, Pn, showed a preference for the selfish option over altruistic and prosocial options, and this tendency was maintained across experiments. Pn's behavior in combination with that of the two mothers from the other two pairs led to an overall increase in the proportion of selfish vs. prosocial options. However, it should be noted that not all mothers showed higher proportions than their offspring, also reflected in the greater dissimilarity among individuals in P-S than P-A trials. Pn chose selfish when selfish was an option, and prosocial when selfish was not an option; she never chose the altruistic option. In a previous study, the same individual failed to help a partner in the absence of any request, while all other individuals tested did so (*Yamamoto, Humle & Tanaka, 2012*). There is one clear difference in the life history of Pn compared to other participants: Pn was hand-raised by humans. If food is always provided by human caretakers, there is no dependence on other chimpanzees, hence sharing food or begging for food from other chimpanzees may be unnecessary. Previous studies showed the opposite pattern, however when having a human as mediator (*Warneken & Tomasello, 2006*). Given our small sample size this explanation is speculative. However, it raises the interesting possibility that the tendencies to share food (prosocial) or provide food to other (altruistic) are not genetically predetermined behavioral traits; instead, they could arise from a gene-environment interaction (*Plomin, DeFries & Loehlin, 1977*). Further studies are required to examine the effect of chimpanzee rearing history on prosocial and altruistic tendencies. One offspring participant showed an increasing trend toward choosing selfish over altruistic options (Am). This individual was an 11-year-old male who at the time of the study was involved in competition with the alpha-male of the group. This social circumstance might indicate a switching point for Am from offspring behavior to more adult-like behavior.

In summary, while it is valid to question (*Skoyles, 2011*) a 60% advantage for prosocial above selfish options (*Horner et al., 2011a*), we found prosocial responses at much higher rates with increasing task experience. This factor could explain the differences found in *Horner et al. (2011a)*. Sampling alternative options to confirm the game's contingencies (*Horner et al., 2011a*) did not occur. Notwithstanding the small sample size, based on our results we suggest that the rank-relationship between partners, in contrast to *Horner et al. (2011a)*, and supporting other authors (*Melis, Schneider & Tomasello, 2011*; *Yamamoto, Humle & Tanaka, 2012*) may modulate prosocial tendencies: with increasing social pressure and hence fear of repercussions from their mothers, female offspring showed altruistic behavior.

Overall, this study confirms that chimpanzees are not "indifferent to the welfare of others" (*Silk et al., 2005*), however their choices reveal a balanced interplay of rationally maximizing their own gains (*Jensen, Call & Tomasello, 2007*) while circumventing repercussions from the partner (*De Waal, 1989*). Further, we provide a new framework for examining social cognition in a computer-guided testing procedure, allowing better identification of effect-modulating factors.

## CONCLUSION

We provide a new framework for accessing prosociality in non-human primates, through the utilization of a controlled computer apparatus. This improvement of the old paradigm allows us to increase the number of trials and prevents the direct participation of humans in the task that could be a distracter or bias in the chimpanzees' choices. Additionally, the touchscreen methodology developed in the study helps control for the effect of visible food along with the ability to increase trial numbers (*Cronin, 2012*).

Our study revealed a preferential tendency towards acting prosocially by chimpanzees when they are faced with two other options: being selfish or altruistic by benefiting themselves or the other, respectively. Ultimately, we hypothesize that the rearing history of chimpanzees and the rank-relationship between partners influenced their positive or negative response towards prosociality.

## ACKNOWLEDGEMENTS

We thank Dr. James R. Anderson for his comments on the earlier versions of this manuscript, Dr. Masaki Tomonaga and the staff of the Language and Intelligence Section for their help and useful comments. We kindly thank the Center for Human Evolution Modelling Research at the Primate Research Institute for daily care of the chimpanzees. We also appreciate the valuable input by Dr. Lydia Hopper and three anonymous reviewers.

### Funding

This research was supported by Grant-in-Aid for Specially Promoted Research: No. 24000001, (PI: Tetsuro Matsuzawa) by the Ministry of Education, Science, Sports and Culture, Japan to both Tetsuro Matsuzawa and Ikuma Adachi: Grant-in-Aids for Scientific Research (S): No. 16H06283, (PI: Masaki Tomonaga), and for Young Scientists (B): No. 22700270 to Ikuma Adachi, the Japan Society for the Promotion of Science (JSPS) to Christoph D. Dahl. The funders had no role in study design, data collection and analysis, decision to publish, or preparation of the manuscript.

### Grant Disclosures

The following grant information was disclosed by the authors:
Grant-in-Aid for Specially Promoted Research: 24000001.
Ministry of Education, Science, Sports and Culture, Japan.
Grant-in-Aids for Scientific Research: 16H06283, 22700270.
Japan Society for the Promotion of Science (JSPS).

### Competing Interests

The authors declare there are no competing interests.

## Author Contributions

- Renata S Mendonça and Christoph D. Dahl conceived and designed the experiments, performed the experiments, analyzed the data, prepared figures and/or tables, authored or reviewed drafts of the paper, approved the final draft.
- Susana Carvalho and Tetsuro Matsuzawa authored or reviewed drafts of the paper, approved the final draft.
- Ikuma Adachi conceived and designed the experiments, performed the experiments, analyzed the data, authored or reviewed drafts of the paper, approved the final draft.

## Animal Ethics

The following information was supplied relating to ethical approvals (i.e., approving body and any reference numbers):

All experiments were carried out in accordance with the 2002 version of the Guidelines for the Care and Use of Laboratory Primates by the Primate Research Institute, Kyoto University. The experimental protocol was approved by the Animal Welfare and Care Committee of the same institute (protocol# 2012-090).

## Data Availability

The raw data are provided in Data S1.

## Supplemental Information

Supplemental information for this article can be found online at http://dx.doi.org/10.7717/peerj.5315#supplemental-information.

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
