# Peer review of "Touch-screen-guided task reveals a prosocial choice tendency by chimpanzees (Pan troglodytes)"

_PeerJ, doi:10.7717/peerj.5315_

## Round 0.1 · original submission · Major Revisions

· Academic Editor

Major Revisions

In your study you applied a robust methodological technique (touchscreens) to test chimpanzees’ prosocial behavior. I was particularly interested to see that, not only had you tested the chimpanzees via a touchscreen interface, but that you had also included conditions via which you could directly compare the chimpanzees’ prosocial and altruistic behavior. Furthermore, touchscreens afford the opportunity to remove physical (food) rewards from tests of decision making, which have been shown to influence primates’ responses in previous tests of prosociality (see, e.g., the responses of the capuchins tested by Claidiere et al. 2015).

Your study, therefore, represents a novel and innovative strategy and one which I think is well suited for publication in PeerJ. However, as the manuscript currently stands, I cannot accept it for publication without some rather substantial changes. Three experts have provided thoughtful and detailed reviews and I encourage you to respond to each of their points to help improve the clarity of your article. In addition to the reviewers’ feedback, I have some points of my own that I ask you address (some echo concerns raised by the reviewers, others are novel).

General comments:

As noted by reviewer 2, given that you tested two chimpanzees in a single testing booth, did they ever share or steal each other’s rewards? I am thinking that this is likely, especially given the mother-offspring relationship. Please report the number of trials in which a subject ate both their own reward, as well as the other subject’s reward (or vice versa). If such stealing did occur, this would explain the chimpanzees’ proclivity to select symbols on the touchscreen that delivered more food rewards. Therefore, without the inclusion of these data, it is difficult to fully interpret the findings.

It is typical in tests of prosociality to have ‘probe’ conditions in which the social partner is absent in order to differentiate whether the donor is simply selecting the option that garners more rewards, or the option that provides rewards to their partner. This concern of mine echoes concerns also raised by reviewer 3. Can you run a control in which either the partner is absent, or is in an adjacent testing booth, to test this? At a minimum, please provide a rationale as to why you did not run such a control.

Additionally, and as noted by all three reviewers, the language throughout contained many grammatical errors, typos and awkward turns of phrase. As PeerJ does not proof edit articles before publication, I strongly encourage you to thoroughly proof read your article to correct this. (Reviewer 2 even provides an annotated copy of your article with some detailed suggestions.)

Introduction:

I absolutely applaud the novelty of using touchscreens to test chimpanzees’ prosocial responses. However, you also claim that, aside from the study by Claidiere et al. (2015), no other study has tested chimpanzees’ prosocial behavior without a barrier between the subjects. This is not the case. Please see House et al 2014 (PLoS ONE, 9(9): e103422) and Suchak et al 2014 (PeerJ, 2:e4147).

Methods:

Like the reviewers, I found the description of the methods difficult to interpret. Please clarify these descriptions throughout. Please also provide a more detailed description of what the symbols represent where you introduce them in your methods (i.e. line 137) and also please report whether the meanings of the symbols were counterbalanced across the three pairs of subjects tested.

Additionally, when you describe your subjects, please state explicitly which chimpanzees were tested together as pairs. I am familiar with your naming system, such that an offspring’s name start with the same letter as their mother’s, but for other readers, this might not be clear.

Results:

Given the inclusion of the three experiments, and without the provision of clear hypotheses as to what each would show (I concur with reviewer 3 that you must include your hypotheses for each experiment), the results section was very difficult to parse out. I also agree with the reviewers that you could probably reduce the number of figures and tables. More generally, this section might benefit from a restructuring of the entire article, such that you present each experimental method and results section in turn e.g., Experiment 1 methods, Experiment 1 results, Experiment 2 methods, Experiment 2 results etc.

For Fig 3, why do you present the cumulative mean? I suggest simply presenting the raw proportion for each bin of trials.

Line 207 – you state “These two-option trials were randomly interspersed with three-option trials at a ratio of 1:5, to ensure that chimpanzees could associate this new condition with the previous one.” Please explain (in your intro or methods section) why you did this. Both, why you felt it was necessary for the chimpanzees to associate experiment 2 with experiment 1 and how you thought such a methodology would achieve this. Additionally, why did you not analyze their responses in these trials (line 210)?

Specific/minor comments:

In some places in your article you refer to the chimpanzees with their full names and in other places you use abbreviations, please be consistent throughout.

Line 84 – when describing the results of Horner et al (2010) you note “given the frequent selection (40%) of selfish tokens.” I know what this is referring to as I am very familiar with the study by Horner et al. However, without such knowledge, this statement would be confusing as you have not described their paradigm (i.e. that they used tokens). Please rephrase this section.

Line 92 – you write “Unlike most of the studies”. I presume this is referring to previously-run studies of prosociality, but the way it is phrased makes it seem like you are referring to the previous cognitive testing experiments that your subjects participated in.

Line 129 – you write “3 to 4 degrees of visual angle” but I do not know what this means. Please clarify.

Throughout your results section you use the phrase “individual testing showed…”. This makes it sound like you were testing subjects individually. Rather, I suggest you use the phrase “individual’s responses showed…”

Line 263 – but see Massen et al. 2011 (Primates, 52:237–247) for a study with macaques which did reveal the role of rank in tests of prosociality

Line 284 – you state “Although rank turned out to be an important factor in our study, because we tested only mother-offspring pairs”. I do not believe you can say this. You did not test for rank.

Reviewer 1 ·

Basic reporting

Context:
From my perspective it seems like all important background is covered. Not really sure why the section on understanding intentions is included in the intro, if you think it is very relevant, could you please clarify, otherwise, maybe this is an opportunity to shorten the MS.

Language:
Please proofread and double-check grammar throughout manuscript. Check lines 75, 91, 98, 99, 102-105, 127, 136, 153, 155, 238, 298, and Table 1, among others.

Tables and Figures:
Could you please add the information about mother-offspring relationships in the tables. I know that this is also explained in the text, but it might help readers understand the table more quickly, too.
Figure 4 the label and caption don’t match (Fig 4 / 6), at least in the PDF that I got.
Figures 4 and 5 include information on “Proportion of choice X above choice Y”, not only for Experiment 2 and 3, but also for Experiment 1. However, in Experiment 1, three choices were provided. So what does the proportion mean? Is “prosocial above altruistic” e.g. “proportion of prosocial choices among all of the choices that were either prosocial or altruistic, ignoring selfish choices”. If that’s what you did, it is difficult to compare to proportions in pure two option choices. At the very least the readers should know. So could you please explain, either in text or in figure caption. Sorry if I misunderstood!
Figures 3, 6 and 7 speak of fitted values but I could not find any information on the underlying models that were fitted. Could you please provide this information in supplementary.
Figures 4, 6 and 7: given that the main message of the MS is that prosocial was chosen over selfish very often, would it not make more sense to flip around the lines from “selfish above prosocial” to “prosocial over selfish” for more intuitive readability?
Figure 5, the caption speaks of error bars, but there are no error bars in the figure
Finally, what is the difference between Figure 4, bottom right, and Figure 5, other than that one seems to be the invert of the other?

Raw Data:
I really appreciate that the authors make available their data, thank you for this!
The data could be presented in a more accessible fashion, here are some suggestions.
The paper speaks of Experiments, but the data speaks of Conditions, please harmonize.
The three conditions (experiments) are currently saved in three different data sheets, but they could be in one big data set that simply has another variable “Experiment” that codes where the data comes from.
It is not clear to me why for the different conditions, some variables have to be repeated in multiple columns (e.g. Condition 1, why are there three columns each for the Individual and Trial variable, and why are there six empty columns). The same is true for Experiments 2 and 3. For all three sub datasets, there could simply be one simple structure “Experiment – Individual – Trial – Prosocial – Altruistic – Selfish”. For trials that provided only two options (in Experiments 2 and 3), the third option has to be a missing value of course.

Experimental design

Description of Methods:
I found it a bit difficult to follow the sections on the training phase (151 - 163). The way I understand the training phase, there was only ever one individual, the individuals did not touch anything, they just heard the three sounds: one for selfish (food to their feeder), one for altruistic (food to what would later become the recipient feeder) and one for prosocial (food to one of the feeders, randomly chosen). And the actor always had access to both feeders, including the recipient feeder. And in the learning phase the screen was active and there was only ever one symbol, the individual touched it and the appropriate feeder delivered a reward. Is this correct? It would be great if this section could clarify this.

Validity of the findings

I think many sensible precautions were taken to avoid that individuals may simply learn to prefer the prosocial option, which, as I understand it, may have been a problem with some earlier studies. The very quick development of clear preferences (Figure 3), and the consistency in the other Experiments speaks for a more “insightful” task understanding than gradual preference learning.

I would recommend to be a bit more careful in making general statements about chimpanzees in the discussion and conclusion (e.g. l 331 “a clear prosocial tendency on chimpanzees”). Your sample is too small for meaningful statistical inference to “a population” of chimpanzees, and even if it was larger, it is not representative of “the chimpanzee” because the PRI chimps are a captive group with many unique features. This is not to say that anything is wrong with these chimpanzees or that their behavior is “unnatural”, I would simply caution not too make too broad a generalization.

The only thing I found a bit confusing was your take on the social pressure / fear of harassment explanation of prosocial behavior. As you point out, one may argue that individuals may behave prosocially when in the same compartment only to avoid being pressured by their partner (l 267 to 268, see also Tennie, Jensen, & Call, 2017 for a recent example). You argue that this may not apply to your own study because of (1) the results by Claidiere et al. speak against it, and (2) you did not observe signs of frustration in your own study (L. 270 to 272).
I wonder if Claidiere et al (2015) is really a good counter-example, because in that study the chimps were trained by the experimenters to always accept the reward distributions, even if a lower ranking individual received more (“In the together condition [… we] removed the apparatus from reach if one or the other tried to reach for the reward on the other side.”).
Further, with regard to signs of frustration in your own study, do you mean you did not see signs of frustration during the early parts of Experiment 1 when individuals still chose selfishly sometimes? Because in later sessions most subjects chose prosocial most of the time (when it was an option) so there would be no reason for either chimp to get frustrated, because, one might argue, their partner had already “preemptively” provided them with food most of the time? Or do you mean that you did not observe frustration or attempts to steal food in the selfish vs. altruistic trials only, where you would most likely expect them to occur (l 224 to 226)?
Finally, when speculating about some of the deviations in your data (why adults choose selfish over altruistic most of the time but two of the offspring choose altruistically occasionally and one does not, l 312-314, 322/3), you say that social pressure may be part of the explanation after all. So why would the offspring choose so un-selfishly (for one individual in Experiment 3 even more often than not), if not because of fear of repercussions? And so if social pressure may play a role in motivating “altruistic” behavior in the altruistic vs. selfish condition, at least by lower ranking individuals, why would this logic not also apply in the prosocial vs selfish condition? I think the behavior still fits your own definition of prosocial in that it is intended to benefit one another, so I am not recommending to rewrite the whole discussion. But it would be good if you could clarify the logic a little bit here. I apologize if I misunderstood.

Additional comments

I think this is a very interesting study and a well written manuscript.

Reviewer 2 ·

Basic reporting

The use of the English language in this manuscript should be improved so that an international audience can clearly understand the text. I have annotated the manuscript to correct grammatical errors and typos. Also throughout the text, many of the numbers used should be spelled out according to Peer J's guidelines.
The authors provide a good introduction to the prosocial literature and context for their study. The literature referenced is relevant as well.
The figures are high quality but there are some issues with the descriptions (please see annotated manuscript).
One broad suggestion is to include an age class column in Tables 2-4 so that it’s easy to see who is an adult and who is juvenile while interpreting the results.

Experimental design

This research is interesting and novel in the field of prosociality with the addition of the touchscreen used in the experiment.
There are some ambiguities in the methodology which I have addressed in the annotated manuscript and below. This was mainly related to the description of experiment 3 and how it differed from experiment 2. To clarify this difference I would also suggest rewriting the description in Table 1 to something like this:
We ran 96 trials for each pair of choices (P-S, S-A, P-A). Each pair was presented in a block of 48 trials and the order of choices counterbalanced between individuals.

In the training phase, were the different sounds also paired with each reward location? The phrasing in lines 157-158 is unclear as it appears that the experimenter was just randomly choosing where to reward, was it a random presentation of sounds paired with the experimenter delivering food to the appropriate locations? Or just random reward locations?
Line 166- was it random whether the adult or juvenile began as actor?
Line 172- did the sound they were trained with also play when they made a choice along with the sound of the feeder?
Lines 174- Chimpanzees only changed positions (recipient vs actor) after an entire experiment was over? As I understand it there were 3 sessions per experiment so I suggest clarifying that after the completion of each experiment the roles were switched. And again in line 175- two experiments were run per day if I interpreted this correctly.
I also think that lines 204-213 should be in the methods section.

Validity of the findings

These findings are significant to the field and the data is analyzed well. I would like to see any data or descriptions about whether there was scrounging of rewards between the chimpanzees during the study. The text states that there weren't behaviors directed towards the partner, but the behavior related to the food could be explicitly noted.
Line 341- I would suggest adding that the touchscreen methodology helps control for the effect of visible food along with the ability to increase trial numbers could cite Cronin 2012 in Animal Behavior.

Additional comments

I think this is an interesting advancement in prosocial testing with the addition of a touchscreen apparatus. With some clarifications regarding methodology and improvements in the language it deserves to be published.

Annotated reviews are not available for download in order to protect the identity of reviewers who chose to remain anonymous.

Reviewer 3 ·

Basic reporting

1. Language: The English is most of the time very good, easy to follow and appropriate for the context. There are some spelling mistakes (e.g. Line 102: to examined; Line 155: so that the actor could easily ear the sound) or wrong wordings (e.g. Line 98: were modulated across different the conditions), which can be easily eliminated by thorough proof reading.
2. Literature: The literature cited is appropriate and informative.
3. Structure: The structure of the manuscript mainly conforms to PeerJs standards. However, no clear hypotheses and predictions are given. Furthermore, the results section includes too many information on methods, which needs to be moved to the Material and Method sections (Lines 189-190; 204-213; 224-228).
4. Figures: There are too many figures and tables showing similar or even the same data (e.g. the right Panel of Figure 4 shows data included in the tables; same goes for Figure 5). Furthermore, the right and left panels of Figure 4 basically show the same thing, the error bars already indicate whether there are large or small differences between individuals. So the right Panel is not really necessary. And if I’d like to know the individual performances, I can look them up in the Tables. I would suggest to better summarize the results in fewer figures and tables, e.g. combine Table 3 and 4. To indicate Mother/Offspring you could just highlight which Initial refers to which category, e.g. typing the mothers in bold, or add a column indicating whether the subject is mother or offspring. And restrict yourself to one sort of diagram for presenting the data (not two as in Figure 4). Furthermore, I do not think that you need to show the cumulative means for all conditions. Figure 7 e.g. shows that not much is happening throughout the course of the experiment, so why do you show us this diagram?
5. Figures: Furthermore, the Figure captions are not sufficient to understand the Figures easily and do show quite some mistakes (spelling and form). E.g. in Figure 4 you should reference each part of the figure with an additional letter (a,b,c..), and explain accordingly. Don’t put the error bar explanation in the middle of the caption. And pay attention that the caption does not start with referencing another Figure (e.g. Figure 4: Figure 6 Mean proportion of choices for S-A trials…). The caption of Figure 3 says “Proportion of the cumulative mean […] as a function of time/trial (x-axis)”. There is no time referenced. Just the number of trials. So please clarify the description. Furthermore, rephrase the ‘bin’ explanations: “Trials are grouped in 8 bins of 18 trials”. You probably mean something like “trials are grouped in bins (each comprising 8 trials) for a total of 144 trials, i.e. 18 bins”.
6. Especially the Figures (3,6,7) showing the cumulative means need more explanations and better referencing. It took me some time to understand the “altruistic above prosocial” (and similar) phrasing in the Figures and Tables. I would suggest rephrasing these descriptions to better understand which data are shown. You mean the results in trials where only the referenced two options had been shown, right (e.g. altruistic and prosocial)? You need to either better explain that in the captions or use a different term.
7. Results: The results sections needs the most improvement. As there are too many Figures and Tables, the referencing is too complicated. After restricting the figures and tables, please rephrase the results section accordingly and explain better. Especially the results of Experiment 1 are difficult to understand. How can you report a “selfish over prosocial” choice (e.g. Line 188, 189), when there always have been 3 options, i.e. selfish, prosocial and altruistic? I do not get that at all. Same goes for reporting these kind of data in Figures 4 and 5. I do not understand how you calculated the “selfish over xy” for Experiment 1. That does not make any sense to me.
8. Raw data are supplied

Experimental design

The research question is indeed interesting and meaningful. However, I think that the experimental design and analysis need some improvement.
1. First, the authors state that they want to better understand the cognitive processes involved in the prosocial choice tasks. However, I do not see how the current experiments examined any cognitive processes? As stated in the discussion the experiments rather explored the influence of “nurturing history and rank-relationships”. So the initial aim of exploring ‘cognitive processes of PCT’ should be reconsidered.
2. Furthermore, the authors state that experiment 2 served as a control to ensure that the chimpanzees understand the task. I do not see that. Why proves a consistent choice in Experiment 2 that the chimpanzees understood the meaning of the task? They could have just learned to choose a specific symbol in Experiment 1, without a deep understanding of all the consequences. Some subjects may have just learned to choose the circle, irrelevant of the second option. Or could have learned to avoid a specific symbol. Actually, Cl ALWAYS choose the prosocial symbol. She may have just learned that this specific symbol is rewarding her. How do you know that she understood the other two symbols? Especially, when forced to choose between the altruistic and selfish option, she chose randomly. Demonstrating that she did not know what the other two symbols meant. I know, you report a learning and training phase at the beginning, but what happened there exactly? No results of these phases are reported. So, the alternative explanations are just missing. I am not convinced that the subjects indeed understood the meaning of each symbol, at least not all of them. And this needs to be discussed.
3. The authors try to explain in the discussion why they did not include a ‘ghost control’. However, I think this is a necessary control condition, rather than Experiment 2. Why can’t the authors just block the subjects access to the ‘partners’ reward, by building a barrier or something? This would solve the issue of over-rewarding the subject in prosocial choices.

Validity of the findings

As stated above, I am not entirely convinced that every chimpanzee understood the meaning of each symbol. Alternative explanations are possible and need to be discussed or even explored further. Therefore, I am not sure that the experiment indeed showed clear prosocial choices in chimpanzees. I feel the authors’ assumption is not valid and at least needs a lot more discussion.

Additional comments

The authors state in the discussion that there had not been any negative interactions between the subjects. Really? So when e.g. Pn always chose the selfish option, his mother just sat there and did nothing? That is hard to believe. Did the experimenters interact with the second chimpanzee somehow? More information on the behaviour of the recipients would be great.

---

## Round 0.2 · Minor Revisions

· Academic Editor

Minor Revisions

Thank you very much for responding to the three reviewers' comments in relation to your original submission as well as those I provided. All three reviewers have reviewed your revised manuscript and all are in agreement that you have made considerable improvements to your manuscript and that, pending some final edits, your manuscript is suitable for publication in PeerJ. I concur.

Both reviewers 1 and 3 note that the language of your article is still somewhat unclear in places. Given their concerns, as well as the fact that PeerJ does not provide proofing, I have taken the liberty to provide edits to your manuscript to help improve the clarity of your language throughout. Please see the attached document showing my suggested tracked changes.

In addition to the suggested edits I have made to your article, I note that there are still some concerns regarding your figure captions. Please double check these before you resubmit. For example, and as noted by reviewer 3, the Fig 6 caption refers to experiment 2. Should this be experiment 3? Additionally, you provide abbreviations in the caption for Fig 3 ("Proportion of the cumulative mean in the experiment 1 for S (selfish), A (altruistic) or P (prosocial) options as a function of trials (x-axis) for five individuals (A) and Pan (B).") that are not referenced in the figure and so I think can be removed (i.e. "Proportion of the cumulative mean in the experiment 1 for selfish, altruistic or prosocial options as a function of trials (x-axis) for five individuals (A) and Pan (B).")

If you can make these final few edits to the language of your manuscript, it will be my pleasure to accept your article for publication in PeerJ.

In addition to the concerns regarding the clarity of the language, I had one comment concerning your conclusions. In your discussion you note that the chimpanzees' increased tendency to act prosocially (as compared to other studies) might be due to the fact that you tested the chimpanzees in the same enclosure. This is indeed likely. However, you should consider recognizing that Claidiere et al 2014 found that chimpanzees tested in the same enclosure were less likely to act prosocially than pairs tested separately in adjacent enclosures.

Finally, while reviewer 3 requests that you reformat your raw data, provided as a supplemental file, I do not believe this is necessary and I am happy for you to use the document you have currently provided without further edits.

Reviewer 1 ·

Basic reporting

-

Experimental design

-

Validity of the findings

-

Additional comments

All my concerns from review round 1 have been addressed to my satisfaction. The English language in the manuscript still needs improvement. I strongly recommend to have a native speaker have a second look at it.

Reviewer 2 ·

Basic reporting

The authors addressed all issues to my satisfaction.

Experimental design

The authors addressed all issues to my satisfaction.

Validity of the findings

The authors addressed all issues to my satisfaction.

Reviewer 3 ·

Basic reporting

Thanks for the revision of your manuscript. I am happy with most of the changes, but have some minor points:

- Thanks for including predictions, however, I still think that the underlying hypotheses need also to be formulated, e.g. why should the prosocial tendency increase over trials? Furthermore, the predictions sound a bit circular/obvious, the way you formulate them. I would suggest to rearrange a little, so it is rather something like: 1) If chimpanzees have a tendency to behave prosocial, they should choose the prosocial option more often than the selfish and altruistic options. In addition, there are again some spelling/grammar erros in this sections, e.g. ll. 113: this tendency increase(s) over trials. Please check again

ll. 174: “didn’t” should be “did not”
ll. 177: This sentence sounds strange/wrong. First, it should be “This way ‘we encouraged’ or ‘the subjects were encouraged’. But ‘encouraged to understand’ is also odd. I would suggest something like: “This procedure should ensure that the individuals understand that both…”
ll. 196: I guess you mean ‘rewarding the ACTOR” not the author
ll. 235: delete ‘the’
ll. 309-311: this sentence needs some rearrangement
ll. 312: it should be ‘have shown’
ll. 327: haven’t should be have not
ll. 330: delete ‘the’ before ‘experiment 3’
Figure 6: In the caption you refer to experiment 2, but Fig. 6 shows the results of experiment 3, right?


Raw Data: The authors copied the results of the experiments next to each other, but that was not what Reviewer 1 intended (at least I think so). Instead of putting the results next to each other, they should have put them underneath and added the column ‘Experiment’, filling in 1, 2 and 3 accordingly. So the results could be analysed much easier. The way it is presented now is not that appropriate.

Experimental design

s. above

Validity of the findings

s.above

---

## Round 0.3 · accepted · Accept

· Academic Editor

Accept

Thank you for responding to the final round of reviewer comments, as well as the editorial suggestions I provided.

It is my absolute pleasure to accept your article for publication in PeerJ.

#